# Dietary Taurine Intake Affects the Growth Performance, Lipid Composition, and Antioxidant Defense of Juvenile Ivory Shell (*Babylonia areolata*)

**DOI:** 10.3390/ani13162592

**Published:** 2023-08-11

**Authors:** Yunchao Sun, Xiangyu Du, Yi Yang, Aimin Wang, Zhifeng Gu, Chunsheng Liu

**Affiliations:** 1Sanya Nanfan Research Institute, Hainan University, Sanya 572025, China; yiyyiziwei1@outlook.com (Y.S.); duxiangyu1104@163.com (X.D.); yiyangouc@outlook.com (Y.Y.); hnugu@163.com (Z.G.); 2College of Marine Biology and Fisheries, Hainan University, Haikou 570228, China; aiminwang@163.com

**Keywords:** ivory shell *Babylonia areolata*, taurine supplementation, growth performance, lipid composition, antioxidant ability

## Abstract

**Simple Summary:**

As a growth-promoting additive, taurine has various physiological functions in aquatic animals. However, few studies have explored the effects of taurine intake on gastropod species’ growth performance and other physiological metabolism. The present study evaluates the effects of different taurine supplementation levels (0.0% as control, 1.0%, 1.5%, 2.0%, 2.5%, and 3.0%) on the growth performance, lipid composition, and antioxidant ability of juvenile ivory shells *Babylonia areolata* and obtains the optimum concentration of taurine for ivory shell culture. The results showed that (1) moderate taurine supplementation in an artificial diet (about 1.5–2.0%) could significantly improve the growth performance and antioxidant ability of ivory shells; (2) taurine supplementation could change the lipid composition of juvenile ivory shells; and (3) taurine supplementation could increase the expression of *orexin* and *Neuropeptide Y* (*NPY*) and inhibit the expression of *leptin* and *cholecyatoklnin*.

**Abstract:**

In this study, an eight-week feeding trial was performed to investigate the effects of different taurine supplementation levels (0.0% as control, 1.0%, 1.5%, 2.0%, 2.5%, and 3.0%) on the growth performance, lipid composition, and antioxidant ability in juvenile ivory shells *Babylonia areolata*. The results showed that taurine supplementation significantly improved the specific growth rates (SGRs) and survival rates of ivory shell (except the survival rate in the 3.0% taurine diet group) (*p* < 0.05). The SGRs showed an increasing and then decreasing tendency with increasing dietary taurine supplementation, and the highest value was observed in the 2.0% taurine diet (2.60%/d). The taurine content in the muscle of ivory shells fed taurine-supplemented diets significantly increased when compared to the control group (*p* < 0.05). The profiles of C22:2n6 in the muscle of ivory shells fed taurine-supplemented diets were significantly higher than in the control group (*p* < 0.05), and the highest values were observed in the 2.0% taurine supplementation group. The high-density lipoprotein cholesterol (HDL-C) content in the hepatopancreas showed an increasing and then decreasing tendency with increasing dietary taurine supplementation, while the low-density lipoprotein cholesterol (LDL-C) concentration showed a decreasing tendency. Furthermore, the activities of pepsin and lipase in both the intestine and hepatopancreas significantly increased at moderate taurine supplementation levels compared to the control group (*p* < 0.05). Accordingly, obvious increases in the histological parameters in the intestine of ivory shells fed taurine-supplemented diets were also found. As for the antioxidant ability, the activities of the total antioxidant capacity (T-AOC) and superoxide dismutase (SOD) showed an increasing and then decreasing tendency with increasing dietary taurine supplementation, and the highest values were observed in the 1.0% and 1.0–2.0% taurine supplementation groups, respectively; the malondialdehyde (MDA) contents significantly decreased with increasing dietary taurine supplementation (*p* < 0.05). The taurine intake affected the expression of four appetite-related genes in the hepatopancreas, in which *orexin* and *NPY* showed an increasing and then decreasing tendency, while *leptin* and *cholecyatoklnin* decreased with increasing dietary taurine supplementation. In conclusion, moderate taurine supplementation in an artificial diet (about 1.5–2.0%) could improve the growth performance and antioxidant ability and change the lipid composition of juvenile ivory shells.

## 1. Introduction

The ivory shell (*Babylonia areolata*), a carnivorous gastropod species of the phylum Mollusca, the class Gastropoda, the order Stenoglossa, and the family Buccinidae, is widely distributed in the tropical and subtropical sea areas of the Indo-Pacific Ocean [1,2]. As an important fishery species, ivory shell has been widely cultured in many East and Southeast Asia countries, such as China, Vietnam, Thailand, etc. [3,4,5]. In China, ivory shells are typically cultured in flow-through concrete/canvas ponds and fed with forage fish (such as *Decapterus maruadsi*) [2,6]. Forage fish are expensive, deteriorate the water quality, and also result in the spread of disease, which limits the sustainable development of the ivory shell industry [7,8]. To resolve these problems, the nutritional requirements and artificial compound feed of ivory shells have been studied, and the optimum protein, lipid, and carbohydrate requirements have been confirmed [7,9,10]. However, in large-scale farming, relatively low growth rates of ivory shells fed dry pellets were observed because of the poor feeding attraction activity of these artificial diets. Our preliminary experiment showed that taurine supplementation could significantly improve the appetite of ivory shells (data unpublished); therefore, we inferred that taurine could be used as feed attractant in artificial diets of ivory shells.

Aquatic animal feed attractants are a small group of chemicals (such as l-amino acids, nucleosides, nucleotides, and glycine betaine) or some marine extracts (such as stick water and squid paste), which could improve the feed intake, ingestion, and growth performance of animals [11,12,13,14,15]. As for marine mollusks, in common with other animals, many studies have revealed that some kinds of chemicals have a strong attraction for certain species. For example, the feeding of black abalone *Haliotis discus* was significantly increased by an ether fraction of allspice *Pimentu officinulis*, specific purine and pyrimidine compounds, and some kinds of amino acids (such as glycine, proline, arginine, and taurine) [12,16,17]; montiporic acids C (1) and A (2) isolated from the coral *Montipora* sp. were also shown to have potent feeding-attractant activity for a coral-eating gastropod *Drupella cornus* [18]; glycine could obviously provoke feeding behavior in the carnivorous opisthobranch sea slug *Pleurobranchaea japonica* [19]. To our knowledge, there have been no reports on the feed attractants of ivory shells.

Taurine (2-aminoethanesulfonic acid) is a non-protein amino acid in the form of free amino acids (FAAs), which has various physiological functions in aquatic animals, such as promoting feeding, antioxidation and immunomodulation, regulating osmotic pressure, and maintaining homeostasis [15,20,21,22]. Until now, the effectiveness of taurine as a feed attractant has been reported in many aquatic species. Koven et al. [23] reported that supplementing taurine (1.1–2.2%) significantly improved the wet weight gain in juvenile white grouper *Epinephelus Aeneus*, and effected the constituent of fatty acids in the liver and eye; Liu et al. [24] showed that an artificial diet with 10 g/kg-taurine was necessary for juvenile turbot *Scophthalmus maximus*, which could enhance the growth rate, serum thyroid hormone level, and antioxidative ability, while an excessive supplement of dietary taurine (100 g/kg) caused an obvious impairment in the liver and distal intestine; Dong et al. [25] also reported that a suitable taurine supplement (0.4–0.8%) was important for improving the growth, regulating the immunity, and enhancing the antioxidant capacity in Chinese mitten crab *Eriocheir sinensis*. Yue et al. [26] showed that white shrimp *Litopenaeus vannamei* fed 0.4–0.8 g/kg taurine-supplemented diets presented a significantly higher weight gain, protein efficiency ratio, and protein retention efficiency. As a common feed attractant, taurine could excite the olfactory/gustatory organs of aquatic animals and promote their feeding [27], improve the intestinal nutrient digestion and absorption [28], maintain the optimal health of aquatic animals [25], and reduce the protein degradation in the liver and muscle [29]. Therefore, taurine has been considered as an important functional amino acid in aquatic animals’ nutrition [21,30].

Though the feed attractant of taurine has been widely studied and has shown potential value in improving the growth rate and disease resistance in fish, shrimp, and crab [30], as far as we know, there are few reports about the effects of taurine intake on gastropod species’ growth performance and other physiological metabolism. Therefore, the purpose of this study was to evaluate the influences of different taurine supplementation levels on the growth performance, lipid composition, and antioxidant ability of juvenile ivory shell and to determine the optimum concentration of taurine for ivory shell culture. Moreover, the feed attractant mechanism of taurine on ivory shells was also explored.

## 2. Materials and Methods

### 2.1. Experimental Diets

In this study, soft artificial feed was used as the basic formula (the ingredients, proximate composition, and taurine content are shown in Appendix A; the fatty acid profiles are shown in Appendix A), which proved to be successful in ivory shell culture (data unpublished). Six experimental diets were supplemented with extra taurine (≥98% chemical purity, Jiakangyuan, Beijing, China) at the levels of 0.0% (as control group), 1.0%, 1.5%, 2.0%, 2.5%, and 3.0% (% weight of the total basal diet). Then, all ingredients were ground with a mill (MJ-PB8G2-071), sifted through an 80-mesh sieve, mixed with 20% clear water, and homogenized. At last, the soft diets were sealed to sausage-like shapes (diameter ~30 mm) with a plastic bag before being stored at −20 °C.

### 2.2. Ivory Shell Rearing, Experimental Design, and Sampling

Juvenile ivory shells (shell height of 21.88 ± 0.51 mm; body weight of 1.71 ± 0.11 g) were obtained from a local commercial hatchery (Wenchang, Hainan Province, China) and acclimatized in 54 cm × 42 cm × 30 cm tanks for one week before experimental manipulation, according to the method of Zhou et al. [2]. After acclimatization, 630 ivory shells with a similar size that could well accept the artificial soft feed were selected for the experimental trial. In detail, the ivory shells were randomly divided into six groups, 105 individuals for each group. For each taurine level group, the ivory shells were randomly divided into three subgroups. Each subgroup included 35 individuals distributed in a 60 L blue square tank, with the bottom covered with 6 cm thick layer of sand and filled with 40 L seawater. The ivory shells were hand-fed with artificial diets daily at 16:00 at approximately 5% of body weight. Feed consumption was recorded for each tank every day. The ivory shells were bulk weighed and counted every 2 weeks to adjust the feeding rate. The water temperature (28.26 ± 1.28 °C), dissolved oxygen (>5.0 mg‧L^−1^), pH (8.11 ± 0.12), and salinity (31.52 ± 0.87) were maintained during the experiment. The tanks were thoroughly cleaned, and the sand was changed biweekly.

At the end of the growth trial, the ivory shells were sacrificed 24 h after the last feeding. The shell height, whole body weight, soft tissue weight, and muscle weight of each ivory shell in each tank were measured. The hepatopancreas and intestines of nine ivory shells in each tank were randomly sampled, three of which were fixed with Bouin’s solution for histological analysis, three were stored at −80 °C for the detection of relevant enzymes, and three hepatopancreas were quickly frozen with liquid nitrogen and stored at −80 °C for gene expression analysis. Meanwhile, the muscles of ivory shells in each tank were collected and stored at −80 °C for subsequent chemical analysis.

### 2.3. Growth Performance and Biological Parameters Analysis

Data were analyzed for the weight gain rate (WGR), specific growth rate (SGR), feed conversion rate (FCR), survival rate (SR), viscera index (VSI), muscle tissue index (MTI), and soft tissue index (STI) using the following formulas:WGR (%) = 100 × [(final weight − initial weight)/initial weight].
SGR (%/d) = 100 × (*ln* final body weight − *ln* initial body weight)/days of experiment.
FCR = amount of feed given (g)/weight gain (g).
SR (%) = (final number of ivory shells/initial number of ivory shells) × 100.
VSI (%) = (viscera weight/whole body weight) × 100.
MTI (%) = (muscle weight/whole body weight) × 100.
STI (%) = (soft tissue weight/whole body weight) × 100.

### 2.4. Histology of Intestine

After more than 24 h of fixation, the intestines were dehydrated in gradient series of ethanol (75%, 85%, 95% and 100%), transferred to xylene for transparency, and then embedded in paraffin. By using an ultramicrotome, solid wax blocks were cut into slices (5 μm) by a microtome (Leica RM2016, Leica, Wetzlar, Germany) and stained with hematoxylin and eosin. These sections were examined under a light microscope (Olympus CX-23, Olympus, Tokyo, Japan) to determine the dimensions of the intestinal fold height (hF), intestinal fold width (wF), and enterocytes (hE) in mid-enteric sections. In addition, the five largest intestinal folds per ivory shell were measured.

### 2.5. Proximate Composition and Free Taurine Concentration Analysis of Muscle

The proximate compositions of ivory shell and diet were analyzed as described by Zhou et al. [2] and Liu et al. [31]. The moisture was measured by drying the sample at 60 °C until a constant weight was obtained; the crude protein content was determined by the Kjeldahl method using a fully automatic Kjeldahl Nitrogen/Protein Analyzer (FOSS-2003, FOSS, Höganäs, Sweden) after acid digestion; the crude lipid was extracted with petroleum ether (FOSS-Soxtec 2050, FOSS, Höganäs, Sweden); the crude ash was determined by incineration in a muffle furnace at 550 °C for 24 h; the glycogen content was determined using the Glycogen Content Kit (colorimetry, Jiancheng Biological Engineering Institute, Nanjing, China), following the manufacturer’s instructions. All samples were analyzed in triplicate.

Taurine analysis of the ivory shell muscle and diet was performed according to the previous research [32]. Each sample was weighed and mixed with three volumes of pre-cold 10% trichloroacetic acid (TCA). After being homogenized, the sample was centrifuged at 4 °C, 10,000× *g* for 15 min, and the supernatant was repeatedly washed with diethyl ether to remove the TCA. Then, the pH of the sample was adjusted to 2.2, filtered, and kept at 4 °C. The analysis of taurine was performed with high-performance liquid chromatography (HPLC) in a Waters 2996 (Waters Corporation, Milford, MA, USA) equipped with a Waters Pico-Tag-C18 column (3.9 mm × 150 mm). All the samples were analyzed in triplicate.

### 2.6. Fatty Acid Analysis

The fatty acids of the ivory shell muscle and diet were extracted with chloroform–methanol (2:1, *v*/*v*), according to the method described in Liu et al. [33]. In detail, the total lipids were extracted from 2 g of homogenized sample with chloroform–methanol (2:1, *v*/*v*), then saponified, followed by esterification, and finally the extraction of fatty acid methyl esters (FAMEs) in hexane. Then, the FAMEs were analyzed using a Gas Chromatography-Mass Spectrometer (GC-MS, 7894A-5975C; Agilent, Santa Clara, CA, USA). Identification of fatty acids was made after a comparison of their retention times with standards (Supelco 37 Component FAME Mix, Bellefonte, PA, USA).

### 2.7. Cholesterol Analysis

The sample (0.15 g) was homogenized in nine volumes of physiological saline under ice-water bath conditions. Then, the homogenized samples were centrifuged at 2500 rpm for 10 min at 4 °C, and the supernatant was collected in a 1.5 mL Eppendorf tube for cholesterol analysis. The protein concentration of each sample was determined using the Coomassie Brilliant Blue protein assay kit (A045-2-2). The total cholesterol (COD-PAP method, A111-1-1), triglyceride (GPO-PAP enzymatic method, A110-1-1), low-density lipoprotein cholesterol (LDL-C, A113-1-1), and light-density lipoprotein cholesterol (HDL-C, A112-1-1) of the hepatopancreas were measured using the relative kits (Jiancheng Biological Engineering Institute, Nanjing, China), according to the manufacturer’s instructions. The cholesterol contents of each diet treatment were analyzed more than five times.

### 2.8. Evaluation of Intestinal and Hepatopancreas Digestive Enzyme Activities

The frozen tissues of ivory shells were weighed, thawed at 4 °C, and homogenized (8000 rpm, 30 s) with 0.9% pre-cooled and sterilized normal saline at a ratio of 1:9 (tissue: saline). These samples were centrifuged at 2500 rpm for 10 min, and the supernatant was collected in a 1.5 mL Eppendorf tube for digestive enzyme activity analysis. The protein concentration of each sample was determined using the Coomassie Brilliant Blue protein assay kit (A045–2-2). The activities of pepsin (colorimetry), amylase (Iodine-starch colorimetry), and lipase (colorimetry) were determined using the respective kits following the manufacturer’s instructions (Jiancheng Biological Engineering Institute, Nanjing, China).

### 2.9. Activity Assay of the Hepatopancreas Antioxidant Enzyme

The pretreatment of the hepatopancreas antioxidant enzyme was the same as the digestive enzyme activities’ detection, and the antioxidant enzymes were detected using commercial kits following the manufacturer’s instructions (Jiancheng Biological Engineering Institute, Nanjing, China). The protein concentration of each sample was determined using the Coomassie Brilliant Blue protein assay kit (A045-2-2). The activities of the total antioxidant capacity (T-AOC, ABTS method, A015-2-1), superoxide dismutase (SOD, WST-1 method, A001-3-2), catalase (CAT, visible light, A007-1-1), and malondialdehyde (MDA, TBA method, A003-1-2) contents were measured via spectrophotometric analysis using a microplate reader.

### 2.10. Gene Expression Analysis

To examine the expression level of the nutrition metabolism-regulation-related genes in hepatopancreas, four appetite-related genes were detected (Appendix A). In detail, RNA extraction and cDNA synthesis were performed as described in a previous study [34]. qRT-PCR was carried out in an ABI 7300 Real-time Detection System (Applied Biosystems, Foster City, CA, USA) using the SYBR ExScript qRT-PCR Kit (Takara, Dalian, China) with β-actin as an internal reference [35]. The assay was conducted in three replicates, and the data were analyzed by the 2^−ΔΔCt^ method [36].

### 2.11. Statistical Analysis

The values are expressed as mean ± standard deviation (S.D.). All statistical analyses were performed using Data Processing System (DPS) statistical software (DPS V18.10) [37]. The data of the growth performance, biochemical indicators, histological measurements, digestive and stress-related enzymes, and expression of the metabolism-regulation-related genes were analyzed using one-way analysis of variance (ANOVA), followed by the least significant difference (LSD) test. Differences were considered statistically significant at *p* < 0.05.

## 3. Results

### 3.1. Growth Performance

As shown in Table 1, the survival rates of the ivory shells in all groups were more than 90% after the eight-week culture, and significant differences were observed in the 1.0–2.5% taurine diet groups compared to the 0.0% and 3.0% groups (*p* < 0.05). The final body weight (FBW), WGR, and SGR of the ivory shells showed an increasing and then decreasing tendency with increasing diet taurine supplementation, in which the ivory shells fed with a 2% taurine diet were the highest. The FCR of the ivory shells fed with a 2% taurine diet was 1.24, which was significantly lower than that of the 0.0% and 3.0% taurine diet groups (*p* < 0.05). There were no significant differences in the MTI, STI, and VSI among the different groups (*p* > 0.05).

### 3.2. Intestinal Morphology

The photomicrographs and histological measurements of intestinal tissues are shown in Figure 1. The sections of the ivory shell intestines of all groups showed no lesions and loss of epidermal integrity, with an intact mucosa and microvillus folds. As for the measurements of the intestinal fold height (hF), intestinal fold width (wF), and enterocyte height (hE), the 2.0% taurine supplementation group had the highest values (138.23 μm, 45.28 μm, and 22.49 μm, respectively), which were significantly higher than the other groups (*p* < 0.05), except for the wF in the 1.5% taurine supplementation group.

### 3.3. Body Biochemical Composition

The contents of crude protein, crude lipid, ash, and glycogen in ivory shell muscle showed an increasing and then decreasing tendency, and the highest values of lipid and ash were observed in the 2.0% taurine supplementation group, while the highest values of crude protein and glycogen were found in the 2.5% taurine supplementation group (Table 2). Furthermore, the taurine content obviously increased with increasing dietary taurine supplementation, in which significant differences were observed in the 2.0%, 2.5%, and 3.0% taurine supplementation groups vs. the 1.5% group vs. the 1.0% group vs. the 0.0% group (*p* < 0.05).

As shown in Table 3, the proportion of polyunsaturated fatty acid (PUFA) was 60.44–63.54%, in which C22:6n3 (docosahexaenoic acid, DHA, 14.76–15.85%), C20:4n6 (arachidonic acid, ARA, 13.60–15.16%), and C20:5n3 (eicosapentaenoic acid, EPA, 7.47–9.37%) were the top three PUFA. In detail, the ARA and EPA profiles in the control group were significantly higher than those in the taurine supplementation groups (*p* < 0.05), while the DHA profile in the 1.5% taurine supplementation group was the highest. The Σn-3 PUFA profile of the control group was significantly higher than in the other groups (*p* < 0.05), while the profile of the Σn-6 PUFA showed the opposite results (*p* < 0.05), which was largely caused by the C22:2n6 changes. Furthermore, the proportion of monounsaturated fatty acid (MUFA) showed an increasing and then decreasing tendency, and the highest value was observed in the 2.0% taurine supplementation group, while the opposite results were observed in the proportion of saturated fatty acid (SFA).

As shown in Table 4, the concentrations of total cholesterol, LDL-C, and HDL-C in the hepatopancreas were affected by taurine supplementation. The total cholesterol contents in the taurine treatments were significantly lower than in the control group (*p* < 0.05), except for the 3.0% taurine supplementation group. The total cholesterol and HDL-C contents in the hepatopancreas showed an increasing and then decreasing tendency, while the LDL-C significantly decreased with the increase in the taurine supplementation, and significant differences were observed in the control, 1.0%, and 1.5% taurine supplementation groups vs. the 2.0%, 2.5%, and 3.0% taurine supplementation groups (*p* < 0.05).

### 3.4. Digestive Enzyme Activity of Intestine and Hepatopancreas

The activities of three digestive enzymes in the intestine and hepatopancreas are shown in Table 5. Generally, the pepsin and lipase activities were higher in the hepatopancreas compared to those in intestine, while the amylase activity showed the opposite tendency. In the intestine, the activities of the three digestive enzymes showed an increasing and then decreasing tendency, and the highest values were found in the 1.5% taurine supplementation group for pepsin and amylase activities, respectively, and in the 1.0% taurine supplementation group for lipase activity. In the hepatopancreas, the pepsin activity of the 2.0% taurine supplementation group was significantly higher than in other groups (*p* < 0.05). The amylase activity of the control group was significantly higher than that of the taurine supplementation groups (*p* < 0.05), while the significantly lowest lipase activity was detected in the control group compared to the other groups (*p* < 0.05).

### 3.5. Oxidative Stress-Related Enzyme Activity of Hepatopancreas

As shown in Figure 2, significant differences of the activities of T-AOC and SOD and the MDA contents were found among the different diet groups (*p* < 0.05). The activities of the T-AOC and SOD showed an increasing and then decreasing tendency with increasing taurine content, and the highest values were observed in the 1.0% and the 1.0–2.0% taurine supplementation groups, respectively, compared to the other groups (*p* < 0.05); the MDA contents significantly decreased with increasing dietary taurine supplementation (*p* < 0.05). Furthermore, there were no significant differences in the CAT among the ivory shells fed different diets.

### 3.6. Gene Expression Related to the Nutrition Metabolism in Hepatopancreas

Figure 3 shows the expression of appetite-related genes in the hepatopancreas of ivory shell. In detail, the mRNA expression levels of *orexin* and *NPY* first increased and then decreased with increasing dietary taurine supplementation, and significant differences were observed in some of the taurine diet groups compared to the control group (*p* < 0.05); the expression levels of *leptin* and *cholecyatoklnin* decreased with increasing dietary taurine supplementation, and their expression levels in the 2.5% and 3.0% taurine supplementation groups were significantly reduced compared to that in the control group (*p* < 0.05).

## 4. Discussion

In the present study, the SRGs of ivory shell were 2.29–2.60%/d during an eight-week growth trial. In a previous study, Zhou et al. [2] reported that the SRGs of juvenile ivory shell (~2.28 g) fed trash fish were no more than 1.04%/d; Chi et al. [38] showed that the SRGs of 5.05 g ivory shells were 0.8–2.1%/d. The relatively higher SRG meant that the artificial diet used in our experiment was suitable for juvenile ivory shells. Furthermore, the SRGs of ivory shells showed an increasing and then decreasing tendency with the increase in the taurine supply, and the highest value was observed in the 2.0% taurine supplementation group. These results indicated that moderate taurine supplementation could significantly increase the growth performance of ivory shells. In previous studies, the optimal dietary taurine contents of aquatic species, even the same species, have been shown to vary because of genetics, life stages, feed formula, and other culture experimental conditions [21] Presumably, high-trophic carnivorous aquatic species have a higher taurine requirement compared to low- and medium-trophic herbivorous species, whose optimum taurine demands were assessed to be 0.61% and 1.36%, respectively [39]. Taurine deficiency might inhibit the growth of aquatic animals, whereas an overdose of dietary taurine also caused intestinal injury and hepatic pathological change, leading to growth inhibition [24,30]. Accordingly, the 1.5–2.0% taurine supplementation in daily diet for ivory shell is reasonable.

Until now, dietary taurine supplementation has been shown to significantly increase the growth rates in many other aquatic animals, including fish (such as white grouper *E. Aeneus* [23] and turbot *S. maximus* [24]), shrimp (white shrimp *L. vannamei* [26]), crab (Chinese mitten crab *E. sinensis* [25]), and sea cucumber *Apostichopus japonicus* [40]. There are two possible reasons by which taurine promotes growth performance: (1) taurine can improve the efficiency of feed; (2) taurine is a recognized dietary attractant that can promote feeding [15,41]. In our experiment, the FCR of the ivory shell fed the 2.0% taurine diet was significantly lower than the control group (*p* < 0.05), which indicated taurine could improve the efficiency of feed in ivory shell. Furthermore, the mRNA expression levels of four appetite-related genes (*leptin*, *orexin*, *NPY*, and *cholecystokinin*) in ivory shell hepatopancreas were detected. In fish species, *leptin* and *cholecystokinin* are two anorexigenic relative genes, while *orexin* and *NPY* are orexigenic relative genes [42,43,44]. In the present study, feeding a taurine diet could stimulate the overexpression of *orexin* and *NPY* and lead to repression of the *leptin* and *cholecystokinin* expression in hepatopancreas. The expression trends of the appetite-related genes indicated that taurine supplement in the artificial diet could promote feeding, which caused the higher growth performance of ivory shell.

The intestine and hepatopancreas are two main digestive organs in mollusks. In the present study, the amylase activities in both the intestine and hepatopancreas of carnivorous ivory shell were lower compared to other herbivorous mollusks, such as strawberry conch *Strombus luhuanus* (~0.5–1.2 U/mg prot in intestine) and pearl oysters *Pinctada fucata martensii* (0.35–1.57 U/mg prot in visceral mass) [45,46]. As for pepsin and lipase, their activities significantly increased in both the intestine and hepatopancreas of ivory shell when fed a moderate taurine-supplemented diet. Similar results were also observed in common carp *Cyprinus carpio* and black carp *Mylopharyngodon piceus* [47,48]. Enterocyte height, intestinal fold height, and width are three important parameters that widely used in evaluating intestine function [15,45]. The present study showed that moderate taurine supplementation in an artificial diet (1.5–2.0%) significantly increased these three parameters in the intestine compared to the control group, which indicated that dietary taurine supplementation had positive effects on the intestine tissues of ivory shell. In a previous study, Liu et al. [24] showed that an excessive level of dietary taurine (100 g/kg) caused impairment in the intestine of turbot *S. maximus*. In our experiment, the highest dietary taurine supplementation was 3.0% of the artificial diet, which was much less than the dietary taurine level in turbot. Accordingly, no obvious histological damages were observed in the intestine of 3.0% taurine treated ivory shells. Similarly, in juvenile snakehead fish *Channa striata*, dietary taurine supplementation (0.5–1.5%) also had no harmful effects on its distal intestine tissues [49].

Studies in aquatic animals have shown that taurine is involved in the synthesis of bile salts, which are essential for the emulsion, digestion, and absorption of dietary lipids [20,50,51]. Lipids, including phospholipids and essential fatty acids, are essential nutrients which significantly affect the growth and development of aquatic animals [15,52,53]. In this study, the lipid contents in the muscle of ivory shells were 1.11–1.28% of wet weight, and C22:6n3 (DHA), C20:4n6 (ARA), and C20:5n3 (EPA) were the top three PUFA, which showed similar results to previous reports [5,32]. As for the lipid changes, taurine supplement increased the crude lipid contents, which showed the opposite tendency with cultured fish Atlantic salmon *Salmo sala* [52]. It is known that cultured fish are superfatted (for example 8.3–9.1% of wet weight lipid content in Atlantic salmon) caused by a relatively high-fat diet or culture condition [52]. Therefore, taurine could reduce the lipid deposition and slightly decrease the lipid content to a normal level [15]. However, the lipid content in ivory shell was much lower; moreover, the lipid content of the ivory shell fed the artificial diet in our study was slightly lower than that fed trash fish (1.11–1.28% vs. 1.62%). Accordingly, taurine might improve the lipid content and result in a lipid increase.

In PUFA, taurine significantly increased the Σn-6 PUFA, which was mostly contributed by C22:2n6 (from 3.53% in the control group to 7.19–10.71% in the taurine supplementation groups) (*p* < 0.05). It is notable that C22:2n6 is one of the major PUFA in many bivalve species, such as oyster (*Crassostrea gigas*, *C. ariakensis*) and clam (*Cerastoderme edule*, *Scrobicularia plana* and *Ruditapes philippinarum*), which share similar features with ivory shell [54,55]. According to a previous study reported by Chen et al. [56], C22:2n6 had potential application as an antitumor nutraceutical supplement and was able to affect the cardiovascular functions. Therefore, the higher C22:2n6 content in ivory shells improved by taurine supplement might have health benefits for humans. As for the cholesterol concentrations’ detection in the hepatopancreas, the LDL-C of the 2.0–3.0% taurine diet groups and the HDL-C of the 1.0–2.0% taurine diet groups significantly decreased and increased, respectively, compared to the control group (*p* < 0.05). The LDL-C is involved in the translocation of lipids from the liver to peripheral tissues, while the HDL-C transports cholesterol from peripheral tissues to the liver for catabolism, which can help to reduce lipid accumulation in aquatic animals [57,58]. The changes in the body lipid of ivory shells might be associated with the changes in the production of LDL-C and HDL-C.

Oxidative stress is an important detoxification mechanism in aquatic animals in response to the fight against pathogenic microorganisms and environmental stress [2,59,60]. T-AOC, SOD, and CAT are three important antioxidative enzymes that reflect the activities of antioxidants in organisms, and malondialdehyde (MDA) is an indicator that reflects the degree of cellular oxidative damage [60,61,62]. In this study, the activities of T-AOC and SOD of the hepatopancreas in the 1.0–2.0% taurine diet groups significantly increased when compared to the control group. Accordingly, the MDA contents also decreased with increasing dietary taurine supplementation. These results indicated that proper taurine supplementation in the daily diet could improve ivory shell’s resistance to various environmental stresses. Taurine has antioxidative properties because of its effect on antioxidative enzymes and genes in the liver of fish [63]. According to previous studies, antioxidative enzymes in freshwater prawn *Macrobrachium nipponense*, sea cucumber *A. japonicus*, and turbot *S. maximus* were significantly increased by dietary taurine supplementation [24,40,64]. Therefore, obviously higher survival rates were observed in the 1.5–2.5% taurine diet groups compared to the control group.

## 5. Conclusions

In conclusion, the growth performance and survival rate of ivory shell were significantly improved when fed a moderate taurine supplementation diet. The crude lipid content and PUFA profile in the 2.0% taurine supplementation groups were the highest. The profile of C22:2n6 in the muscle and the HDL-C content in the hepatopancreas showed an increasing and then decreasing tendency with the increasing dietary taurine supplementation, while the LDL-C concentrations showed a decreasing tendency. Furthermore, moderate taurine supplementation could improve the antioxidant ability (increasing the activities of T-AOC and SOD and decreasing the MDA contents). In the hepatopancreas, taurine supplementation could increase the expression of *orexin* and *NPY* and inhibit the expression of *leptin* and *cholecyatoklnin*.

## Figures and Tables

**Figure 1 animals-13-02592-f001:**
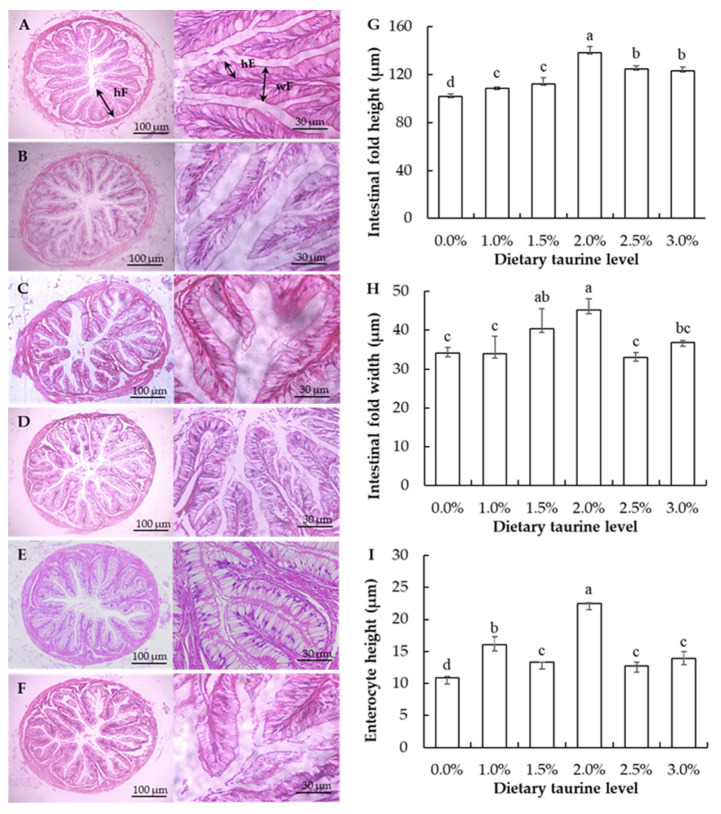
Photomicrographs and histological measurements of the intestine of ivory shells fed artificial diets with different taurine supplementation levels. Note: (**A**–**F**) are photomicrographs of the intestinal tissue sections of 0.0%, 1.0%, 1.5%, 2.0%, 2.5%, and 3.0%, respectively (H&E, 40× and 400×, respectively); (**G**–**I**) are the histological measurements of ivory shell intestines with different treatments. Means ± S.D. of the intestinal morphology parameters are presented. n = 5 replicates per treatment. Different letters of the same intestinal morphology parameters represent significant differences (*p* < 0.05).

**Figure 2 animals-13-02592-f002:**
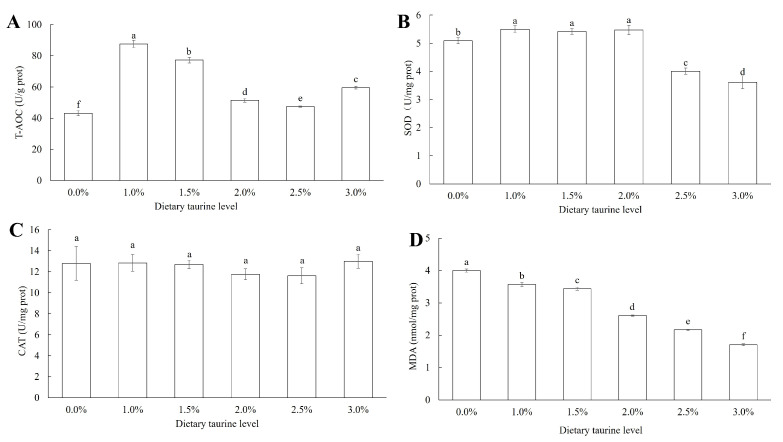
Antioxidant enzyme activities in the hepatopancreas of ivory shell fed artificial diets with different taurine supplementation levels. Note: (**A**–**D**) are activities or contents of T-AOC, SOD, CAT and MDA, respectively. Means ± S.D. are presented. n = 5 replicates per treatment. Different letters represent significant differences (*p* < 0.05).

**Figure 3 animals-13-02592-f003:**
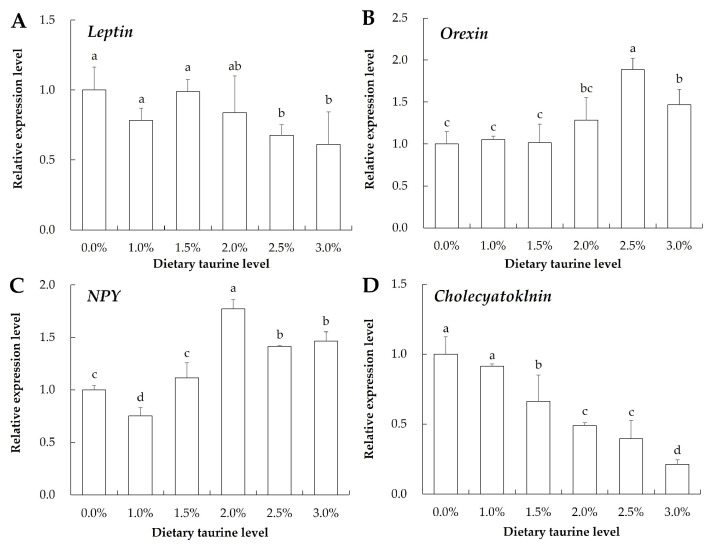
Expression of four appetite related genes in the hepatopancreas of ivory shell fed artificial diets with different taurine supplementation levels. Note: (**A**–**D**) are expression of *Leptin*, *Orexin*, *NPY* and *Cholecyatoklnin*, respectively. Means ± S.D. are presented. n = 5 replicates per treatment. Different letters of the same gene represent significant differences (*p* < 0.05).

**Table 1 animals-13-02592-t001:** The effects of the taurine intake on the growth performance and body mass index of the ivory shells.

	0.0%	1.0%	1.5%	2.0%	2.5%	3.0%
IBW (g)	1.71 ± 0.11					
FBW (g)	6.18 ± 0.05 ^d^	7.00 ± 0.10 ^b^	7.18 ± 0.05 ^a^	7.32 ± 0.07 ^a^	6.68 ± 0.05 ^c^	6.81 ± 0.13 ^c^
WGR (%)	261.40 ± 2.56 ^d^	309.36 ± 5.64 ^b^	319.88 ± 3.18 ^a^	328.07 ± 4.44 ^a^	290.64 ± 2.69 ^c^	298.25 ± 7.31 ^c^
SGR (%/d)	2.29 ± 0.01 ^f^	2.52 ± 0.02 ^c^	2.56 ± 0.012 ^b^	2.60 ± 0.02 ^a^	2.43 ± 0.01 ^e^	2.47 ± 0.03 ^d^
FCR	1.47 ± 0.06 ^ab^	1.38 ± 0.07 ^abc^	1.31 ± 0.08 ^bc^	1.24 ± 0.20 ^c^	1.40 ± 0.03 ^abc^	1.51 ± 0.07 ^a^
SR (%)	90.60 ± 0.06 ^c^	92.89 ± 1.43 ^bc^	98.60 ± 1.43 ^a^	95.46 ± 2.86 ^ab^	98.60 ± 1.43 ^a^	90.03 ± 1.43 ^c^
VSI (%)	12.75 ± 0.97 ^a^	12.37 ± 0.26 ^a^	13.35 ± 0.82 ^a^	13.33 ± 0.51 ^a^	13.43 ± 0.67 ^a^	12.36 ± 1.36 ^a^
MTI (%)	38.08 ± 0.87 ^a^	37.36 ± 1.12 ^a^	37.63 ± 0.93 ^a^	37.41 ± 1.27 ^a^	37.42 ± 0.56 ^a^	38.21 ± 1.45 ^a^
STI (%)	50.83 ± 1.62 ^a^	49.73 ± 1.25 ^a^	50.98 ± 0.36 ^a^	50.74 ± 1.53 ^a^	50.85 ± 1.20 ^a^	50.56 ± 1.70 ^a^

Data represent the means ± S.D. (n = 3). Values in a row with different superscripts (lower case alphabet) indicate significant differences among treatments (*p* < 0.05). IBW: initial body weight; FBW: final body weight; WGR: percent weight gain rate; SGR: specific growth rate; FCR: feed conversion rate; SR: survival rate; VSI: viscera index; MTI: muscle tissue index; STR: soft tissue index.

**Table 2 animals-13-02592-t002:** Proximate composition and taurine content of the muscle (wet weight) of ivory shells fed artificial diets with different taurine supplementation levels.

	0.0%	1.0%	1.5%	2.0%	2.5%	3.0%
Moisture (%)	71.10 ± 0.35 ^a^	69.92 ± 0.54 ^c^	70.69 ± 0.62 ^ab^	70.10 ± 0.20 ^bc^	70.08 ± 0.01 ^bc^	70.65 ± 0.17 ^ab^
Crude protein (%)	17.21 ± 0.27 ^b^	17.27 ± 0.36 ^b^	16.32 ± 0.42 ^c^	17.61 ± 0.14 ^ab^	17.81 ± 0.01 ^a^	16.50 ± 0.11 ^c^
Crude lipid (%)	1.11 ± 0.02 ^c^	1.25 ± 0.03 ^ab^	1.25 ± 0.02 ^ab^	1.28 ± 0.01 ^a^	1.13 ± 0.01 ^c^	1.22 ± 0.02 ^b^
Ash (%)	3.26 ± 0.05 ^c^	3.63 ± 0.08 ^ab^	3.68 ± 0.07 ^a^	3.68 ± 0.07 ^a^	3.47 ± 0.19 ^b^	3.46 ± 0.06 ^b^
Glycogen (mg/g)	3.03 ± 0.09 ^cd^	3.05 ± 0.09 ^c^	3.15 ± 0.01 ^bc^	3.20 ± 0.10 ^b^	3.96 ± 0.01 ^a^	2.92 ± 0.05 ^d^
Taurine (mg/g)	2.00 ± 0.24 ^d^	8.06 ± 0.09 ^c^	8.77 ± 0.09 ^b^	9.45 ± 0.73 ^a^	9.81 ± 0.38 ^a^	10.05 ± 0.05 ^a^

Data represent means ± S.D. (n = 3). Values in a row with different superscripts (lower case alphabets) indicate significant differences among treatments (*p* < 0.05).

**Table 3 animals-13-02592-t003:** Fatty acid profiles (%) of the muscle of ivory shells fed artificial diets with different taurine supplementation levels.

	0.0%	1.0%	1.5%	2.0%	2.5%	3.0%
C14:0	2.65 ± 0.15 ^abc^	2.76 ± 0.03 ^a^	2.50 ± 0.03 ^c^	2.50 ± 0.14 ^c^	2.57 ± 0.12 ^bc^	2.72 ± 0.10 ^ab^
C15:0	0.38 ± 0.07 ^ab^	0.42 ± 0.02 ^a^	0.29 ± 0.11 ^b^	0.34 ± 0.01 ^ab^	0.31 ± 0.11 ^ab^	0.39 ± 0.04 ^ab^
C16:0	16.00 ± 1.35 ^a^	15.79 ± 0.05 ^a^	15.42 ± 0.70 ^a^	12.96 ± 1.35 ^b^	16.08 ± 0.64 ^a^	17.11 ± 1.09 ^a^
C17:0	1.17 ± 0.09 ^ab^	1.05 ± 0.05 ^b^	1.17 ± 0.01 ^ab^	1.04 ± 0.02 ^b^	1.22 ± 0.26 ^ab^	1.27 ± 0.11 ^a^
C18:0	10.44 ± 1.36 ^a^	10.05 ± 0.29 ^a^	8.71 ± 0.81 ^b^	7.67 ± 0.16 ^b^	8.22 ± 0.38 ^b^	8.15 ± 0.40 ^b^
SFA	30.63 ± 3.01 ^a^	30.07 ± 0.25 ^a^	28.10 ± 0.59 ^b^	24.50 ± 1.04 ^c^	28.40 ± 0.76 ^b^	29.64 ± 0.67 ^a^
C16:1n7	0.73 ± 0.27 ^a^	0.58 ± 0.02 ^a^	0.54 ± 0.01 ^a^	0.64 ± 0.11 ^a^	0.52 ± 0.02 ^a^	0.62 ± 0.05 ^a^
C18:1n9 (z)	2.89 ± 0.21 ^c^	2.68 ± 0.12 ^c^	3.46 ± 0.18 ^b^	3.53 ± 0.32 ^ab^	3.34 ± 0.10 ^b^	3.94 ± 0.35 ^a^
C18:1n9 (e)	0.32 ± 0.02 ^b^	0.19 ± 0.03 ^b^	0.34 ± 0.01 ^b^	0.78 ± 0.01 ^a^	0.61 ± 0.36 ^a^	0.27 ± 0.04 ^b^
C20:1n9	3.77 ± 1.71 ^c^	5.34 ± 0.24 ^bc^	5.59 ± 0.22 ^ab^	7.01 ± 1.20 ^a^	5.64 ± 0.47 ^ab^	5.10 ± 0.36 ^bc^
MUFA	7.71 ± 1.67 ^c^	8.79 ± 0.07 ^bc^	9.94 ± 0.05 ^b^	11.96 ± 1.62 ^a^	10.11 ± 0.19 ^b^	9.92 ± 0.03 ^b^
C16:3n3	0.80 ± 0.01 ^cd^	0.71 ± 0.01 ^d^	0.91 ± 0.11 ^bc^	0.92 ± 0.13 ^bc^	1.08 ± 0.11 ^a^	1.01 ± 0.02 ^ab^
C18:2n6	7.39 ± 0.48 ^a^	7.40 ± 0.23 ^a^	6.82 ± 0.07 ^b^	6.86 ± 0.26 ^b^	6.94 ± 0.03 ^ab^	7.07 ± 0.33 ^ab^
C18:3n6	0.19 ± 0.07 ^c^	0.39 ± 0.01 ^b^	0.62 ± 0.08 ^a^	0.47 ± 0.11 ^b^	0.64 ± 0.04 ^a^	ND
C18:3n3	1.07 ± 0.08 ^ab^	1.01 ± 0.01 ^b^	1.23 ± 0.17 ^a^	1.11 ± 0.16 ^ab^	1.09 ± 0.01 ^ab^	1.13 ± 0.05 ^ab^
C20:2n6	1.96 ± 0.16 ^ab^	1.86 ± 0.05 ^ab^	2.00 ± 0.12 ^a^	1.77 ± 0.22 ^b^	1.84 ± 0.15 ^ab^	1.92 ± 0.05 ^ab^
C20:4n6(ARA)	15.16 ± 0.42 ^a^	14.49 ± 0.33 ^b^	14.38 ± 0.08 ^b^	13.63 ± 0.76 ^b^	14.60 ± 0.11 ^b^	13.80 ± 0.40 ^b^
C20:5n3 (EPA)	9.37 ± 0.30 ^a^	8.31 ± 0.05 ^b^	8.73 ± 0.25 ^ab^	7.47 ± 0.96 ^c^	8.27 ± 0.16 ^b^	8.69 ± 0.08 ^ab^
C22:2n6	3.53 ± 0.51 ^c^	7.63 ± 0.09 ^b^	7.19 ± 0.19 ^b^	10.71 ± 3.03 ^a^	7.26 ± 0.50 ^b^	7.63 ± 0.26 ^b^
C22:5n3	6.83 ± 0.35 ^a^	4.57 ± 0.38 ^bc^	4.22 ± 0.51 ^c^	5.36 ± 0.21 ^b^	4.57 ± 0.34 ^bc^	4.35 ± 0.96 ^c^
C22:6n3 (DHA)	15.34 ± 0.49 ^ab^	14.76 ± 0.23 ^b^	15.85 ± 0.24 ^a^	15.24 ± 0.20 ^ab^	15.20 ± 0.30 ^ab^	14.83 ± 0.57 ^b^
PUFA	61.66 ± 1.35 ^b^	61.14 ± 0.18 ^b^	61.96 ± 1.53 ^ab^	63.54 ± 0.57 ^a^	61.49 ± 0.57 ^b^	60.44 ± 0.64 ^b^
Σn-3	33.42 ± 0.24 ^a^	29.37 ± 0.12 ^c^	30.94 ± 1.29 ^b^	30.09 ± 1.26 ^bc^	30.21 ± 0.02 ^bc^	30.01 ± 0.51 ^bc^
Σn-6	28.24 ± 1.58 ^d^	31.77 ± 0.06 ^b^	31.02 ± 0.24 ^bc^	33.45 ± 0.68 ^a^	31.27 ± 0.56 ^bc^	30.42 ± 0.13 ^c^
Σn-3/Σn-6	1.19 ± 0.08 ^a^	0.92 ± 0.01 ^cd^	1.00 ± 0.03 ^b^	0.90 ± 0.06 ^d^	0.97 ± 0.02 ^bcd^	0.99 ± 0.01 ^bc^

Data represent means ± S.D. (n = 3). Values in a row with different superscripts (lower case alphabets) indicate significant differences among treatments (*p* < 0.05). Σn-3: C16:3n3, C18:3n3, C20:5n-3, C22:5n3 and C22:6n3; Σn-6: C18:2n6, C18:3n6, C20:2n6, C20:4n6 and C22:2n6.

**Table 4 animals-13-02592-t004:** Effects of different dietary taurine levels on various cholesterol concentrations (μmol/g prot) of the hepatopancreas of ivory shells fed artificial diets with different taurine supplementation levels.

	0.0%	1.0%	1.5%	2.0%	2.5%	3.0%
Total cholesterol	33.75 ± 2.60 ^a^	28.45 ± 2.00 ^b^	28.92 ± 2.00 ^b^	28.77 ± 1.51 ^b^	23.91 ± 3.14 ^e^	29.86 ± 2.00 ^ab^
Triglyceride	602.15 ± 46.41 ^a^	635.10 ± 11.43 ^a^	648.81 ± 19.37 ^a^	613.73 ± 94.47 ^a^	570.73 ± 49.32 ^a^	655.33 ± 79.78 ^a^
LDL-C	45.78 ± 1.37 ^a^	44.51 ± 8.90 ^a^	42.35 ± 1.71 ^a^	29.15 ± 2.04 ^b^	23.25 ± 2.29 ^b^	27.31 ± 7.39 ^b^
HDL-C	5.24 ± 2.52 ^c^	8.61 ± 0.05 ^ab^	6.84 ± 2.36 ^abc^	9.44 ± 1.84 ^a^	5.68 ± 0.78 ^bc^	5.32 ± 1.44 ^c^

Data represent means ± S.D. (n = 5). Values in a row with different superscripts (lower case alphabets) indicate significant differences among treatments (*p* < 0.05). LDL-C: low-density lipoprotein cholesterol; HDL-C: low-density lipoprotein cholesterol.

**Table 5 animals-13-02592-t005:** Effects of different dietary taurine supplementation levels on the digestive enzyme activities of the intestine and hepatopancreas of ivory shells.

	0.0%	1.0%	1.5%	2.0%	2.5%	3.0%
Intestine						
Pepsin activity (U/mg prot)	6.88 ± 0.07 ^bc^	7.83 ± 0.06 ^b^	10.31 ± 0.13 ^a^	7.40 ± 0.05 ^b^	6.31 ± 0.05 ^c^	6.00 ± 1.16 ^c^
Amylase activity (U/mg prot)	0.16 ± 0.01 ^d^	0.21 ± 0.01 ^b^	0.29 ± 0.02 ^a^	0.20 ± 0.01 ^b^	0.19 ± 0.01 ^bc^	0.17 ± 0.01 ^cd^
Lipase activity (U/g prot)	1.90 ± 0.25 ^b^	2.10 ± 0.23 ^b^	1.89 ± 0.25 ^b^	2.69 ± 0.21 ^a^	1.90 ± 0.21 ^b^	1.48 ± 0.21 ^c^
Hepatopancreas						
Pepsin activity (U/mg prot)	13.73 ± 1.92 ^bc^	12.15 ± 2.72 ^c^	15.75 ± 0.61 ^b^	22.31 ± 0.36 ^a^	13.36 ± 1.79 ^bc^	15.33 ± 2.87 ^bc^
Amylase activity (U/mg prot)	0.03 ± 0.00 ^a^	0.01 ± 0.00 ^c^	0.02 ± 0.00 ^b^	0.02 ± 0.00 ^b^	0.02 ± 0.00 ^b^	0.02 ± 0.00 ^b^
Lipase activity (U/g prot)	2.25 ± 0.07 ^f^	5.97 ± 0.05 ^e^	7.64 ± 0.12 ^c^	6.70 ± 0.34 ^d^	9.24 ± 0.20 ^b^	10.49 ± 0.21 ^a^

Data represent means ± S.D. (n = 5). Values in a row with different superscripts (lower case alphabets) indicate significant differences among treatments (*p* < 0.05).

## Data Availability

Data supporting the findings of this study are available from the corresponding authors (C.L.) on request.

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
