# Peer review of "Dietary Taurine Intake Affects the Growth Performance, Lipid Composition, and Antioxidant Defense of Juvenile Ivory Shell (Babylonia areolata)"

_animals, 2023, doi:10.3390/ani13162592_

Round 1

Reviewer 1 Report

In this manuscript, the effects of dietary taurine intake on growth performance, lipid metabolism and antioxidant defense of juvenile ivory shell were first studied. The results were interesting and meaningful. Some mistakes must be corrected before this manuscript can be accepted.

1. Is there any mechanism that taurine functions in aquatic animals? These information should be included in the second graph of the introduction section.

2. Why was the hepatopancreas used in cholesterol concentrations detection, rather than other tissues in this experiment?

3. Provide more information about the cholesterol analysis.

4. Why the SGR decreased as the concentration of dietary taurine supplementation increased to above 2%? The authors had better to discuss that.

The English should be improved.  Some minor modifications, but not limited to:

Line 22: provide the full name of NPY.

Line 29: “the highest vales” should be “the highest value.”

Line 75: “were also found shown” delete found or shown.

Line 140-147, the abbreviation has been given and it can be directly used in the formula.

Table S2: provide the NCBI number of each detected gene.

Figure 1G: The first letter of Y-axis should be in capital letter.

Line 251: “SD” should be S.D.

Line 322: Figure 2 “SD” should be S.D.

Author Response

Dear reviewer,

Thanks for your comments. We have studied comments carefully and have made correction which we hope to meet with approve. Revised portions are marked in red in the paper. Our response of the comments is enclosed at the end of this letter. Thank you again for your assistance.

Sincerely yours,

Chunsheng Liu

  1. Is there any mechanism that taurine functions in aquatic animals? These information should be included in the second graph of the introduction section.

Response to comments: According to the reviewer’s advice, the mechanism of taurine as feed attractant has been added in the third paragraph 3, line 98-102. In detail, “As a common feed attractant, taurine could excite the olfactory/gustatory organs of aquatic animals and promote their feeding [27]; improve intestinal nutrient digestion and absorption [28]; maintain optimal health of aquatic animals [25]; and reduce protein degradation in the liver and muscle [29]. Therefore, taurine has been considered as an important functional amino acid in aquatic animals’ nutrition [21,30].

  1. Why was the hepatopancreas used in cholesterol concentrations detection, rather than other tissues in this experiment?

Response to comments: Cholesterol, triglyceride and HDL-C in aquatic animal are mainly synthesized by liver/hepatopancreas. LDL-C is transformed from very low-density lipoprotein cholesterol (VLDL-C) in plasma and degraded in liver/hepatopancreas. Cholesterol produced in the liver/hepatopancreas can be transferred to the bloodstream and play an important role in the production of hormones, vitamin D, etc. HDL-C could transfer esterified cholesterol to the liver/hepatopancreas for metabolism. Therefore, liver/hepatopancreas is one of the most important tissues in cholesterol metabolism. Furthermore, the cholesterol concentrations of liver/hepatopancreas were also detected in lipid metabolism of other aquatic animals, such as Chinese mitten crab Eriocheir sinensis (Gao et al, 2020) and red swamp crayfish Procambarus clarkia (Wan et al, 2023).

References:

[1] Wan, W.; Xu, J.; Shi, J.; Zhang, X.; Wang, A.; Dong, X.; Miao, X. The interaction of dietary niacin and lipid affects the growth, antioxidant capacity and lipid metabolism of male Eriocheir sinensis. Aquacult. Rep. 2023, 30, 101541. [https://doi.org/10.1016/j.aqrep.2023.101541]

[2] Gao, F.; Liu, J.; Wang, A.; Liu, B.; Tian, H.; Zheng, X.; Jia, X.; He, C.; Li, X.; Jiang, G.; Chi, C.; Liu, W.; Zhang, D. Dietary lipid sources modulate the intestinal transport of fatty acids in the red swamp crayfish Procambarus clarkii. Aquaculture 2020, 521, 735091. [https://doi.org/10.1016/j.aquaculture.2020.735091]

  1. Provide more information about the cholesterol analysis.

Response to comments: According to the reviewer’s advice, the pretreatment process of cholesterol analysis of hepatopancreas was added in the manuscript. In detail, “Sample (0.15 g) was homogenized in nine volumes of physiological saline under ice-water bath conditions. Then the homogenized samples were centrifuged at 2,500 rpm for 10 min at 4 °C and supernatant was collected in a 1.5 mL eppendorf tube for cholesterol analysis.

  1. Why the SGR decreased as the concentration of dietary taurine supplementation increased to above 2%? The authors had better to discuss that.

Response to comments: Thanks. According to the reviewer’s advice, the disadvantage of overfeeding taurine has been added in the discussion, line 372-374. In detail, “Taurine deficiency might inhibit the growth of aquatic animals, whereas overdose of dietary taurine also caused intestinal injury and hepatic pathological change, leading to growth inhibition.

  1. The English should be improved.  Some minor modifications, but not limited to:

Line 22: provide the full name of NPY.

Line 29: “the highest vales” should be “the highest value.”

Line 75: “were also found shown” delete found or shown.

Line 140-147, the abbreviation has been given and it can be directly used in the formula.

Figure 1G: The first letter of Y-axis should be in capital letter.

Line 251: “SD” should be S.D.

Line 322: Figure 2 “SD” should be S.D.

Response to comments: Thanks. All these mistakes have been corrected. And the grammatical mistakes of whole manuscript have been checked.

  1. Table S2: provide the NCBI number of each detected gene.

Response to comments: Thanks. As for the cDNA sequence of β-actin (Shen et al, 2018), the reference has been added in the manuscript. The cDNA sequence of these four appetite-related genes were obtained from our hepatopancreas transcriptome database, which has not been published. Therefore, they have not been uploaded to NCBI yet. But we can list them as follow:

  • Leptin:

ATGTTTGTCCTGTTCTTCTACGTGCTGTCGCCTGTGCCCACTGTGATCGCCCGGCGCCTGGCCGACAGCCTGGACTCTGCCAGCAGCGCCTGTGTGGAGTTGTGCATTTTCCTCACCACAGGCATCGTCGTTTCCTCCATCGGTCTGCCCATCGTCCTTGCCCACACGCATGTGATTCAGTGGGGCGCCTGCGCTCTGGTCCTTGCAGGCAATGTGGTGGTCTTCTCCACCATCTTGGGGTACTTCCATGTGTTTGGCAGCGATGAGTTTGACTACAGCATGTGGTGA

  • Orexin (partial):

ATGGACGTTCACATCGGCCAGACCAGCTGTTTGGTCTTCAGTCCCAGTCGCCTCCTTAATGGTGACAGCAACCGCCCGTTAAATAACGAACGGCATTCCACTTCTCTTTCCGCAACCTCCTTACCCACCTCACTTACGGACAAAAGCCTTCTTCAATGCCTGAACGACGAAAAGGCTTTTCTCTATATCCCCGTCATCGTCTTCCTGGCCGTGCTGGTCATTGTGGGCACCCTTGGCAACACGCTCGTTCTGTGTGTCTACTGGCGCAAACCATACAAGGCTTCTTCACACTACTTCATCTTATCTCTGGCCGCGCTCGATCTGTTCAGCTGCTTGGTAGGTCTACCCACAGAAATTGCCGATCTGCGCTTTCCTTACCTGTTCGACTTCCCCATCGCCTGCAAACTACTCAGGTTCACCCATTCCAGCACCATCATCGCTTCGAGCAGTATTCTGATTCAGGTTGCCTTTGACCGCTATTATCGCATCTGCAAGCTGGGCCAGCAGTTCAGCGTGAGGAAGGCCAAAATTCTGTGCATCATGTCCATCATCATGGGAATCCTCACCTCCTGGCCCTCCTGCCTTCTCTTTGGTCGCAAGACCCTCAACCTGCACGTACTAACTTCCCCACCTGACGGGCGTTATTACGGCAACCAACAGAGCCAAACTAACGGGATTCACTTTGGTGACAGGGTTCCCATTCAGCGAGAACGGCTGTTGCGGGCCAGTGACTGCTCTACTGAGGATTCTATGCGTGCCACGATCTACCCGACCATCTATTACCTCTTCCTCTTCACGCTCTTCTTCCTCACAGTCACTTTCTTTGCTGTCCTCTACATCCGCATAGGTGTCGCCATTTGGAAACGAAAGCAAAAAACGATCGGGGCCAAGGTCGCCTCCAAGGTCAATAGCCATGGTTCTGACCACACCAGCACAGAAGTCTCCTCCGAGGTGGAAGCAGCAGCTGACAATGCCCATCACCACCCTCCGCCAACCCCTTACCTAACCGGCACTCTGACACGCGGCAGTGGTAAGAAGAAGGAAGCAGAAGCAGAGATGGTGCCGCGACAAAACAAGAGGAGACAAATTCGTGTGGGGAAGACAACTACTGTGCTCTTTGCCGTAACGTTGGCCTATATCCTGAGTTTCCTGCCCTACCTGGTCGTGATGATTTTACGCAGCACTATCAAGGACTTCGAGGGACGTCTTAGTCCTGTGGGAGAGGTGGCCTACAAGTTCTGCGTCAAGTCCTTCTTCATCAACAACGCAATCAACCCCCTCATCTACAGCTTCCTCAATGCCAGCTTCAGAGCTGATACACGAAAAACCCTGCGCAAAATGTGGCACAACTGCTGCTGCCACTGCTTCTGCTGTGAGAGGAACTGTTGCTGTGGAGAGCTTGATTTGGACGTGTTGCCCCAGCAGAGTCCTATTGCTGTG

  • NPY (partial):

ATGCGGCTGACTTGGCTGATGTCACTCTTCATCTTCTTCTACTCGCTCATCTTCCTGCTGGGCACGTTTGGCAACCTGCTGGTGGTGATCGTGGTGCTGCGCAACAAGGCCATGCACACCATCACCAACATCTTCATCACCAACCTGGCCGTGTCGGACATCTGCATGTGCCTGCTCTCCGTGCCCTTCACCCCCATGTCGTTCTTCTTCAACTCCTGGGTCTTCGGCAAGGCGCTGTGCCACGTTGTGCCCATGACGCTCTGCATCAGTGTCTACGTGTCCACCCTCACCTCCACCGCCATCTCCGTC

  • Cholecystokinin

CGGCGAAACTGTTCGAGGAATTAACTCGTTCTCTACAGCTCAGGCGGAAAAGAGACACCGCCACCGCCGCCCCGTTTAACCTGGTCATGATCCCAACCGGCACCAGCTTCCGGCGGAAGAAGAGGGCGGCCTACAGCCGGAAGCGGACGACTTTCATCATGTTCCTCATGACCTTGACCTCCGTGGTGAGTTACCTCCCTTACATCCTGGTCAGCATCGCGTTCGTGGCATCGCGCGGGCTGGATCGCGAGCTGGAAGAAAACGCGGCGGCGAAAGCGGTGGTGGAGGTTTGTCTCAAGTCGTTTCTGATCAGCAGCGCCGTAAACCCGTTCATTTACGGGTTCTGCAATCAGCGGTTCAAGAAGGAGTGTTCCAAGCTGTGGCAAGGGTTCAGCTGTCGTGCGCGTGAGTCTGAGACGGAAGAGACTTCAGAGACTCCGCCTAACAGCTGAGTTATTCTCTGACAGCACTTTTTGGTTGATTAACTCACTCAGTACGGCAAGTCCTCTCT

Reviewer 2 Report

The manuscript presents interesting information for effects of different taurine supplementation on growth performance lipid metabolism and antioxidant defense of juvenile ivory shell (Babylonia areolata). Purpose, materials and methods, results presentation with figures and tables, discussion and references are sound and clear. It is interesting that the fatty acid profiles and cholesterol concentrations of ivory shells would change to by different dietary taurine levels. I have made some comments for better performance. 

Simple Summary

It is OK.

Abstract

Line 26-27, “the specific growth rate (SGR) and survival rate” should be in the plural.

Line 39, “with increasing taurine content” changes to “with increasing dietary taurine supplementation”.

Line 46, add “and” before “change”.

Introduction

Line 62-63, “ivory shells fed dry pellets showed lower growth rate compared to that fed forage fish, because of their high hardness and poor feeding attraction activity.” This sentence is ambiguous.

Line 80-92, introduction of feed attractant mechanism of taurine in aquatic animals should be added in this part.

M & M

How is the concentration range of taurine determined.

Provide the purity of the taurine.

Line 124, replace the “℃” with “ºC”, and unify the units in the text.

Results

Significant difference letters in Table 1 and other Tables should be in the same format (superscript formatting or normal).

The first letter in Figure 1G (histological) needs to be capitalized.

Line 270, “higher than taurine supplementation groups” should be “higher than those in taurine supplementation groups”.

Table S1, there are too many blank spaces in this table.

Discussion

Line 352-354, the square brackets “[]” should be replaced with parentheses “()”.

It is basically accurate, but some expressions still need improvement.

Author Response

Dear reviewer,

Thanks for your comments. We have studied comments carefully and have made correction which we hope to meet with approve. Revised portions are marked in red in the paper. Our response of the comments is enclosed at the end of this letter. Thank you again for your assistance.

Sincerely yours,

Chunsheng Liu

  1. Line 26-27, “the specific growth rate (SGR) and survival rate” should be in the plural.

Response to comments: Thanks. The mistake has been corrected.

  1. Line 39, “with increasing taurine content” changes to “with increasing dietary taurine supplementation”.

Response to comments: According to reviewer’s advice, this part has been changed.

  1. Line 46, add “and” before “change”.

Response to comments: The mistake has been corrected.

  1. Line 62-63, “ivory shells fed dry pellets showed lower growth rate compared to that fed forage fish, because of their high hardness and poor feeding attraction activity.” This sentence is ambiguous.

Response to comments: This sentence has been rewritten as follow “However, in large-scale farming, relatively low growth rates of ivory shells fed dry pellets were observed because of poor feeding attraction activity of these artificial diets”.

  1. Line 80-92, introduction of feed attractant mechanism of taurine in aquatic animals should be added in this part.

Response to comments: According to the reviewer’s advice, feed attractant mechanism of taurine in aquatic animals has been added as follow “As a common feed attractant, taurine could excite the olfactory/gustatory organs of aquatic animals and promote their feeding; improve intestinal nutrient digestion and absorption; maintain optimal health of aquatic animals; and reduce protein degradation in the liver and muscle. Therefore, taurine has been considered as an important functional amino acid in aquatic animals’ nutrition.

  1. How is the concentration range of taurine determined.

Response to comments: The concentration range of taurine used in this study was determined by preliminary experiment. In detail, 11 soft artificial diets with different taurine levels (0%, 0.5%, 1%, 1.5%, 2%, 2.5%, 3%, 3.5%, 4%, 4.5% and 5%) were used. The feed-inducing effect and the 30-min food intake dose were measured. The results showed that 2% taurine supplementation diet group was the best. Therefore, moderate taurine supplementation (1%-3%) and a control (no taurine addition) were tested in the long-time feeding experiment.

  1. Provide the purity of the taurine.

 Response to comments: Thanks. The purity of the taurine was ≥ 98%, and this information has been added in the manuscript.

  1. Line 124, replace the “℃” with “ºC”, and unify the units in the text.

Response to comments: The mistake has been corrected.

  1. Significant difference letters in Table 1 and other Tables should be in the same format (superscript formatting or normal).

 Response to comments: Thanks. The significant difference letters have been uniformed in the manuscript.

  1. The first letter in Figure 1G (histological) needs to be capitalized.

 Response to comments: The mistake has been corrected.

  1. Line 270, “higher than taurine supplementation groups” should be “higher than those in taurine supplementation groups”.

Response to comments: The sentence has been corrected.

  1. Table S1, there are too many blank spaces in this table.

 Response to comments: According to other reviewer’s, more information has been added in this table.

  1. Line 352-354, the square brackets“[]” should be replaced with parentheses “()”.

 Response to comments: The mistake has been corrected.

  1. It is basically accurate, but some expressions still need improvement.

 Response to comments: Thanks. According to the reviewer’s advice, the grammatical mistakes of the whole manuscript have been checked.

Reviewer 3 Report

This paper presents the findings from an experiment to assess the dietary supplementation with taurine on growth performance, lipid metabolism, and antioxidant defense of juvenile ivory shells (Babylonia areolata). The authors demonstrated growth performance and survival rate of ivory shells were significantly improved, as well as crude lipid content, associated to improve the antioxidant ability and expression of orexin and NPY. I believe there is much merit to the data presented in this paper and the findings are very consistent. The findings can help other researchers to understand the beneficial effect of taurine supplementation. However, minor problems were observed in the manuscript, the fact that compromises its publication thereby. I would encourage the authors to adjust the manuscript to publish it in this important journal. In this context, the following comments should be addressed:

- Lines 31 to 33: The authors should rephrase this sentence “…showed an increasing and then decreasing tendency with the increasing dietary taurine supplementation…” it is not suitable.

- Lines 112 to 127: Item 2.2 (methodology), it is not clear the experimental design, the authors described seven groups with 105 juvenile ivory shells each. Then, the experiment should have 735, and not 630 as mentioned (sex groups of 105). At the same time, the authors presented results in only six groups (0.0%, 1.0%, 1.5%, 2.0%, 2.5%, and 3.0%) across the entire study.

- Lines 247 to 253: Figure 1. Authors should share the graphic in three graphics (hF, wF, and hE) for better understanding of readers (the same presented in Figure 2).

- Line 334 to 336: Figure 3. The same, authors should share the graphic in four graphics (one for each gene expression).

Author Response

Dear reviewer,

Thanks for your comments. We have studied comments carefully and have made correction which we hope to meet with approve. Revised portions are marked in red in the paper. Our response of the comments is enclosed at the end of this letter. Thank you again for your assistance.

Sincerely yours,

Chunsheng Liu

  1. Lines 31 to 33: The authors should rephrase this sentence “…showed an increasing and then decreasing tendency with the increasing dietary taurine supplementation…” it is not suitable.

Response to comments: Thanks. According to the reviewer’s advice, this sentence has been rewritten as follow “The profiles of C22:2n6 in muscle of ivory shells fed taurine-supplemented diets were significantly higher than control group (P<0.05), and the highest values were observed in the 2.0% taurine supplementation group. The high-density lipoprotein cholesterol (HDL-C) content in hepatopancreas showed an increasing and then decreasing tendency with the increasing dietary taurine supplementation, while the low-density lipoprotein cholesterol (LDL-C) concentration showed a decreasing tendency.

  1. Lines 112 to 127: Item 2.2 (methodology), it is not clear the experimental design, the authors described seven groups with 105 juvenile ivory shells each. Then, the experiment should have 735, and not 630 as mentioned (sex groups of 105). At the same time, the authors presented results in only six groups (0.0%, 1.0%, 1.5%, 2.0%, 2.5%, and 3.0%) across the entire study.

Response to comments: I am sorry for this mistake. In this experiment, six ivory shell groups were performed in this study. We have corrected this mistake.

  1. Lines 247 to 253: Figure 1. Authors should share the graphic in three graphics (hF, wF, and hE) for better understanding of readers (the same presented in Figure 2).

 Response to comments: According to the reviewer’s advice, the Figure 1G has been divided into three graphics (Figure 1G, H and I).

  1. Line 334 to 336: Figure 3. The same, authors should share the graphic in four graphics (one for each gene expression).

Response to comments: Thanks. Figure 3 has been re-edited.

Reviewer 4 Report

The present study evaluated the effects of dietary taurine supplementation on growth performance, tissue fatty acid and cholesterol contents, and antioxidant defense of juvenile ivory shell. Nutritional information for mollusk is rather lacked. The study demonstrated that dietary taurine supplementation can enhance growth and antioxidant capacity of  ivory shell. There are some drawbacks the authors should address/clarify before publihsing.

1. The authors must reconsidered the term of "lipid metabolism". The authors evaluated fatty acid profile in muscle, and cholesterol and triglyceride concentrations in heppatopancreas. It is hard to discuss lipid metabolism based on these data.

2. L. 84-93, until now, there are many studies focused on taurine nutrition for fish and shrimp. The authors could survey more literature in Introduction section.

3. L. 94, the authors indicated taurine can be used as feed attraction for aquatic animals. However, in the experimental design, the feeding rate was stationary at 5% of body weight. The characteristic of taurine as feed attractant is difficult to be evaluated under this experimental design.  

4. The authors should analyze the actual taurine concentration in all experimental diets.

5. L. 104, the study used soft artificial feed (moist feed, moisture ~50%, Table S1). The authors should provide proximate composition data for all experimental diets, and present the values as % of dry weight. Also, the fatty acid profile for the feeds are required to be given.  

6. L. 118, was it seven groups or six groups?

7. L. 123, the authors stated that "Feed consumption was recorded for each tank every day". Did the authors collected the uneaten feeds? 

8. L. 134, the sentence "muscles of each tank" is confused.

9. L. 139, what is "soft tissue", please define it. It should be noted that the sum of viscera index, muscle tissue index, and soft tissue index was about 100% (Table 1). The values are strange. I don't think the shell weight was so low. The authors MUST recheck the data carefully.

10. L. 159, why the drying temperature so low (60 degree C)?

11. L. 184, please indicate which tissue was analyzed for cholesterol.

12. Table 4, the unit of choesterol and triglyceride concentrations are strange (μmol/g protein). Why did the authors calculate the concentration against protein content? Please recheck it. In addition, why did the authors determine the LDL-C and HDL-C concentrations in hepatopancreas? but not in circulating system (hemolymph)?

13. Fig. 2, the unit of T-AOC activity and MDA content should be rechecked. The antioxidant defense system in animals includes two parts, i.e. enzymatic and non-enzymatic systems. The T-AOC is calculated by total ROS cleavage ability. Furthermore, the MDA is analyzed for fatty acid peroxide metabolites contents. Therefore, the units of μmol/g protein or nmol/mg protein are inadequate.

Overall, the manuscript is acceptable after the authors make appropriate revisions.

Author Response

Dear reviewer,

Thanks for your comments. We have studied comments carefully and have made correction which we hope to meet with approve. Revised portions are marked in red in the paper. Our response of the comments is enclosed at the end of this letter. Thank you again for your assistance.

Sincerely yours,

Chunsheng Liu

  1. The authors must reconsider the term of "lipid metabolism". The authors evaluated fatty acid profile in muscle, and cholesterol and triglyceride concentrations in heppatopancreas. It is hard to discuss lipid metabolism based on these data.

Response to comments: Thanks. According to the reviewer’s advice, the “lipid metabolism” has been replaced by “lipid composition”. Furthermore, the mistakes are also changed in other parts of this manuscript.

  1. L. 84-93, until now, there are many studies focused on taurine nutrition for fish and shrimp. The authors could survey more literature in Introduction section.

Response to comments: According to the reviewer’s advice, more literatures about taurine nutrition for fish and shrimp have been cited in the manuscript. In detail, “Yue et al. [26] showed that white shrimp Litopenaeus vannamei fed 0.4-0.8 g/kg taurine-supplemented diets presented significant higher weight gain, protein efficiency ratio and protein retention efficiency. As a common feed attractant, taurine could excite the olfactory/gustatory organs of aquatic animals and promote their feeding [27]; improve intestinal nutrient digestion and absorption [28]; maintain optimal health of aquatic animals [25]; and reduce protein degradation in the liver and muscle [29]. Therefore, taurine has been considered as an important functional amino acid in aquatic animals’ nutrition [21,30].

References:

[1] Yue, Y.; Liu, Y.; Tian, X.; Gan, L.; Yang, H.; Liang, G.; He, J. The effect of dietary taurine supplementation on growth performance, feed utilization and taurine contents in tissues of juvenile white shrimp (Litopenaeus vannamei, Boone, 1931) fed with low-fishmeal diets. Aquac. Res. 2013, 44, 1317–1325. [https://doi.org/10.1111/j.1365-2109.2012.03135.x]

[2] Kuzmina, V.V., Gavrovskaya, L.K., Ryzhova, O.V.. Taurine. Effect on exotrophia and metabolism in mammals and fish. J. Evol. Biochem. Physiol. 2010, 46, 19–27. [https:// doi.org/10.1134/S0022093010010020]

[3] Rimoldi, S., Finzi, G., Ceccotti, C., Girardello, R., Grimaldi, A., Ascione, C., Terova, G.. Butyrate and taurine exert a mitigating effect on the inflamed distal intestine of European sea bass fed with a high percentage of soybean meal. Fish. Aquat. Sci. 2016, 19, 1–14. []https://doi.org/10.1186/s41240-016-0041-9.

[4] Matias, A.C., Dias, J., Barata, M., Araujo, R.L., Bragança, J., Pous˜ ao-Ferreira, P.. Taurine modulates protein turnover in several tissues of meagre juveniles. Aquaculture 2020, 528, 735478. []https://doi.org/10.1016/j.aquaculture.2020.735478.

  1. L. 94, the authors indicated taurine can be used as feed attraction for aquatic animals. However, in the experimental design, the feeding rate was stationary at 5% of body weight. The characteristic of taurine as feed attractant is difficult to be evaluated under this experimental design.  

Response to comments: According to our preliminary experiment, the daily food intake of ivory shell as no more than 5% of body weight. Therefore, the food intakes were different when ivory shell fed diets with different taurine supplementations. During the 8-week culture experiment, the residual feed of each tank was collected, and the total food intakes of each ivory shell group were calculated, which were used for feed conversion rate (FCR) calculation. As shown in table 1, the total food intakes of ivory shells fed diets with different taurine supplementations significantly higher than that of control group (P<0.05).

Table 1. Total food intakes of ivory shells fed artificial diets with different taurine supplementation levels.

0.0%

1.0%

1.5%

2.0%

2.5%

3.0%

Total food intake (g)

230.07±12.47c

255.52±13.44ab

250.69±12.57ab

243.49±5.66bc

243.49±2.28bc

269.57±13.55a

  1. The authors should analyze the actual taurine concentration in all experimental diets.

Response to comments: As shown in Table S1, the taurine concentration of control diet (basic formula) was 0.61 mg/g. in this experiment, extra taurine (1.0%, 1.5%, 2.0%, 2.5% and 3.0% of the total basal diet) was added to the control diet. The actual taurine concentrations in each experimental diet were 0.61, 10.40, 15.30, 20.20, 25.09 and 29.99 mg/g, respectively.

Table 2. Taurine concentration of each taurine supplemental diet.

0.0%

1.0%

1.5%

2.0%

2.5%

3.0%

Taurine concentration (mg/g)

0.61

10.40

15.30

20.20

25.09

29.99

  1. L. 104, the study used soft artificial feed (moist feed, moisture ~50%, Table S1). The authors should provide proximate composition data for all experimental diets, and present the values as % of dry weight. Also, the fatty acid profile for the feeds are required to be given.  

Response to comments: As described in Question 4, the basic formula was the same in different taurine supplemental diets, except for the taurine concentration. Therefore, proximate compositions of each diet with different taurine supplementations were all most the same as control diets (0.0% taurine supplementations), which has been provided in Table S1. Furthermore, according to the reviewer’s advice, the values (% of dry weight) of soft artificial feed have been added in Table S1. The fatty acid profiles of the basic soft artificial feed are also added as supplementary materials (Table S2).

  1. L. 118, was it seven groups or six groups?

Response to comments: I am sorry. It was six groups, and the mistake has been corrected.

  1. L. 123, the authors stated that "Feed consumption was recorded for each tank every day". Did the authors collected the uneaten feeds? 

Response to comments: Yes. As the description of Question 3, the residual feed of each tank was collected 2 h after feeding. In detail, the wet weight of soft artificial feeds prior to feeding was recorded and the leftover was collected and weighed. Prior to weighing the excess water was removed by tissue paper.

  1. L. 134, the sentence "muscles of each tank" is confused.

Response to comments: The mistake has been corrected as “muscles of ivory shells in each tank”.

  1. L. 139, what is "soft tissue", please define it. It should be noted that the sum of viscera index, muscle tissue index, and soft tissue index was about 100% (Table 1). The values are strange. I don't think the shell weight was so low. The authors MUST recheck the data carefully.

Response to comments: Thanks. In bivalve and gastropod species, soft tissue is the tissue of animal body except for its shell part. In ivory shell, the soft tissue includes viscera and muscle tissues. The whole-body weight is the sum of soft tissue and shell. Furthermore, the abbreviations of viscera index, muscle tissue index, and soft tissue index in “part 2.3”and “Table 1” has been uniformed.

  1. L. 159, why the drying temperature so low (60 degree C)?

Response to comments: As far as we known, the drying temperature for moisture detection is 105 ºC (Liu et al., 2015) or 60 ºC (Li et al., 2020; Zhou et al., 2023). Therefore, 60 ºC was used in our experiment.

References:

[1] Liu, C.; Chen, S.; Zhuang, Z.; Yan, J.; Liu, C.; Cui, H. Potential of utilizing jellyfish as food in culturing Pampus argenteus juveniles. Hydrobiologia 2015, 754(1), 189-200. [https://doi.org/10.1007/s10750-014-1869-6

[2] Zhou, J.; Liu, C.; Yang, Y.; Yang, Y.; Gu, Z.; Wang, A.; Liu, C. Effects of long-term exposure to ammonia on growth performance, immune response, and body biochemical composition of juvenile ivory shell, Babylonia areolate. Aquaculture 2023, 562, 738857. [https://doi.org/10.1016/j.aquaculture.2022.738857]

[3] Li, C.; Liu, E.; Li, T.; Wang, A.; Liu, C.; Gu, Z. Suitability of three seaweeds as feed in culturing strawberry conch Strombus luhuanus. Aquaculture 2020, 519, 734761. [https://doi.org/10.1016/j.aquaculture.2019.734761]

  1. L. 184, please indicate which tissue was analyzed for cholesterol.

Response to comments: Liver/hepatopancreas plays an important role in cholesterol metabolism. Therefore, the concentration of total cholesterol, triglyceride, LDL-C andHDL-C of hepatopancreas were detected.

  1. Table 4, the unit of choesterol and triglyceride concentrations are strange (μmol/g protein). Why did the authors calculate the concentration against protein content? Please recheck it. In addition, why did the authors determine the LDL-C and HDL-C concentrations in hepatopancreas? but not in circulating system (hemolymph)?

Response to comments: HDL-C involved in the translocation of lipids from peripheral tissues to the liver for catabolism can help to reduce lipid accumulation. By contrast, the LDL-C transports cholesterol from the liver to peripheral tissues. Generally, plasma and liver/hepatopancreas are the two mainly target tissues for analysis of LDL-C and HDL-C metabolism in fish (Bai et al., 2021), crayfish (Gao et al., 2020) and crab (Wan et al., 2023). However, it is difficult for juvenile ivory shell (6-7 g) to get enough highly purified plasma compared to fish, shrimp and crab. Therefore, the hepatopancreas were used as cholesterol analysis in this experiment.

In this study, the choesterol and triglyceride concentrations of hepatopancreas were performed according to the relative kits (Jiancheng Biological Engineering Institute, Nanjing, China). Before comparing the differences of choesterol and triglyceride concentration in different taurine supplemented groups, the hepatopancreas samples were homogenized with nine volumes physiological saline, and centrifuged. The supernatant was collected for cholesterol analysis. Then the protein concentration of each supernatant sample was determined using the Coomassie Brilliant Blue protein assay kit. The final value of choesterol and triglyceride were unformed by the soluble protein concentrations. Therefore, the unit of choesterol and triglyceride concentrations was showed as μmol/g protein. Accordingly, the method of cholesterol analysis has been rewritten in revised manuscript.

References:

[1] Wan, W.; Xu, J.; Shi, J.; Zhang, X.; Wang, A.; Dong, X.; Miao, X. The interaction of dietary niacin and lipid affects the growth, antioxidant capacity and lipid metabolism of male Eriocheir sinensis. Aquacult. Rep. 2023, 30, 101541. [https://doi.org/10.1016/j.aqrep.2023.101541]

[2] Gao, F.; Liu, J.; Wang, A.; Liu, B.; Tian, H.; Zheng, X.; Jia, X.; He, C.; Li, X.; Jiang, G.; Chi, C.; Liu, W.; Zhang, D. Dietary lipid sources modulate the intestinal transport of fatty acids in the red swamp crayfish Procambarus clarkii. Aquaculture 2020, 521, 735091. [https://doi.org/10.1016/j.aquaculture.2020.735091]

[3] Bai, F.; Niu, X.; Wang, X.; Ye, J. Growth performance, biochemical composition and expression of lipid metabolism related genes in groupers (Epinephelus coioides) are altered by dietary taurine. Aquacult. Nutr. 2021, 27, 2690–2702. [https://doi.org/10.1111/anu.13395]

  1. Fig. 2, the unit of T-AOC activity and MDA content should be rechecked. The antioxidant defense system in animals includes two parts, i.e. enzymatic and non-enzymatic systems. The T-AOC is calculated by total ROS cleavage ability. Furthermore, the MDA is analyzed for fatty acid peroxide metabolites contents. Therefore, the units of μmol/g protein or nmol/mg protein are inadequate.

Response to comments: Thanks very much. The unit of T-AOC activity should be U/g prot. The Figure 2 has been changed. The content of MDA in hepatopancreas was detected using commercial kits (Jiancheng Biological Engineering Institute, Nanjing, China), which has been reported in many aquatic animals, such as subadult swimming crab Portunus trituberculatus (Long et al., 2019), freshwater prawn Macrobrachium nipponense (Ding et al., 2021) and winged pearl oyster Pteria penguin (Gu et al., 2020). Furthermore, the “/g protein” means the values of MDA content are unformed by the soluble protein concentrations, which is the same as Question 12.

References:

[1] Long, X.; Wu, R.; Wu, X.; Hou, W.; Pan, G.; Zeng, C.; Cheng, Y. Effects of dietary fish oil replacement with blended vegetable oils on growth, lipid metabolism and antioxidant capacity of subadult swimming crab Portunus trituberculatus. Aquacult. Nutr. 2019, 25, 1218–1230. [https://doi.org/10.1111/anu.12936]

[2] Ding, Z.; Kong, Y.; Qi, C.; Liu, Y.; Zhang, Y.; Ye, J. The alleviative effects of taurine supplementation on growth, antioxidant enzyme activities, hepatopancreas morphology and mRNA expression of heat shock proteins in freshwater prawn Macrobrachium nipponense (De Haan) exposed to dietary lead stress. Aquacult. Nutr. 2021, 27, 2195–2204. [https://doi.org/10.1111/anu.13354]

[3] Gu, Z.; Wei, H.; Cheng, F.; Wang, A.; Liu, C. Effects of air exposure time and temperature on physiological energetics and oxidative stress of winged pearl oyster (Pteria penguin). Aquacult. Rep. 2020, 17, 100384. [https://doi.org/10.1016/j.aqrep.2020.100384]

Round 2

Reviewer 4 Report

The authors have made appropriate revisions on their manuscript. It is acceptable for publishing.